

# Climate changes recorded by Hani Peat in Northeast China over the past 13.8 cal ka BP

Ge Shi[a, b, c], Hong Yan[a, b, d] [*], Wenchao Zhang[e], Haobai Fei[a], Shuanshuan Cao[a], Xiaolin Ma[a], Chengcheng Liu[a, c], Fengyan Lu[a], John Dodson[a, f], Henk Heijnis[g], Weijian Zhou[a] and Zhisheng An[a]

[a] State Key Laboratory of Loess and Quaternary Geology, Institute of Earth Environment, Chinese Academy of Sciences, Xi'an, 710061, China
[b] CAS Center for Excellence in Quaternary Science and Global Change, Xi'an, 710061, China.
[c] University of Chinese Academy of Sciences, Beijing, 100049, China
[d] Open Studio for Oceanic-Continental Climate and Environment Changes, Qingdao National Laboratory for Marine Science
and Technology, Qingdao, 266061, China
[e] Key Lab of Submarine Geosciences and Prospecting Techniques, Ministry of Education, and College of Marine Geosciences, Ocean University of China, Qingdao, 266100, China.
[f] School of Earth, Atmospheric and Life Sciences, University of Wollongong, Wollongong, 2500, Australia
[g] Institute for Environmental Research, Australian Nuclear Science and Technology, New South Wales, 2234, Australia

*Correspondence to*: Hong Yan (yanhong@ieecas.cn)

**Abstract.** The Hani peatland is one of the few that remain well-preserved in northeast China, which makes it a valuable site for paleoclimate research. Here, two sediment cores, which cover the past 13.8 ka, were collected, and Loss on Ignition (LOI$_{550℃}$) and X-ray Fluorescence Scanning (XRF) were carried out to build organic matter content and Rb/Sr ratio profiles, in order to assess the climate changes and associated East Asian Summer Monsoon (EASM) evolution since the last

deglaciation. The results show that organic content and the chemical weathering index increased from the early to mid Holocene, possibly reflecting increased precipitation and an enhanced EASM. During the mid to late Holocene, the organic content and the chemical weathering index values decreased, implying that the EASM weakened. The variations of monsoon intensity during the Holocene derived from the Hani peat are consistent with the EASM reconstructions from the Gonghai, Daihai, Qinghai Lake, Hexiazi Island and the Yulin loess-paleosol section. Thus the Hani and other published EASM records

from northern China demonstrate that the evolution of EASM during the Holocene was likely to be dominated by the combination of the influences from changing solar insolation and northern hemisphere ice volumes. In addition, a 0.5-2 ka band filtering analysis of LOI$_{550℃}$ data show that millennial scale climate changes in northeast China were teleconnected with the North Atlantic ice-rafted debris and solar irradiance records, indicating that both North Atlantic climate changes and solar activity probably affected EASM variations.

## 1 Introduction

Variability of the EASM has a significant impact on almost every aspect of East Asian hydrology and ecology, and affects almost one fourth of the world's population. Thus, it is important to study the EASM changes and to identify possible



forcing mechanisms at different temporal and spatial scales (Chen et al., 2015; Liu et al., 2015). Although many studies on the evolution of EASM since the Last deglaciation, some conclusions remain unresolved. For example, stalagmite $\delta^{18}O$ records suggest that the EASM was strongest in the early Holocene (Fleitmann et al., 2003; Wang et al., 2005; Dykoski et al., 2005; Hu et al., 2008; Shao et al., 2006; Wang et al., 2008; Cai et al., 2010; Cheng et al., 2012), but pollen and lake sediment records from the area reveal that the EASM strengthened during the early Holocene and was strongest at the mid Holocene (Xiao et al., 2004; Shen et al., 2005; Chen et al., 2015;Liu et al., 2015;Zhang et al., 2016; Lu et al., 2018). Some early studies indicate that the maximum precipitation had a temporal shift in the EASM area, occurring in the early Holocene for northern China, but in the late Holocene for southeast China (An et al., 2000). However, some recent studies show that the climate was humid during the mid Holocene in both southern and northern China, but arid for central China. (Zhou et al., 2004; Xie et al., 2013; Chen et al., 2016; Rao et al., 2016a, b). These proxies suggest that the regional difference of precipitation shows a diversity of EASM effects during the Holocene. For example, some studies suggested that the stalagmite $\delta^{18}O$ records reflect the variations of atmospheric circulation instead of precipitation (Rao et al., 2014; Chen et al., 2015; Liu et al., 2015). And the reverse pattern of long-term precipitation in northern, southern and central China was proposed to be related to ENSO activity (Xie et al., 2013; Rao et al., 2016a, b).

Peat usually develops in stable sedimentary environments and its properties are sensitive to climate changes. With relatively high temporal resolution, its humification, elemental content, organic content, pollen content, stable isotopes and more are all effective paleoclimate proxies, and are useful for studying past climate changes (Gorham, 1991). In China, many peatlands have been used to reconstruct climate changes across different time scales (Xiao et al., 2004; Hong et al., 2003, 2005, 2009; Zhou et al., 2010; Lu et al., 2013; Chen et al., 2015; Liu et al., 2015; Zhao et al., 2017; Lei et al., 2017; Li et al., 2017; Xu et al., 2006; 2013; Ferret et al., 2012; Zheng et al., 2018). These have greatly improved our understanding of the dynamics of the EASM. Hani peatland in Jilin Province of northeast China is one of the few peatlands that remain well-preserved (C. Schroder et al., 2007). It is situated at the frontal area of the EASM and previous studies have shown that past environmental changes can be reconstructed from pollen, grain size, stable isotopes as well as elemental and organic indicators (Ficken et al., 2000; Zhou et al., 2010; Hong et al., 2005, 2009; Hong et al., 2010; Seki et al., 2009; McClymont et al., 2010; Yamamoto et al., 2010; Zheng et al., 2010, 2011a, b, 2017, 2018; Huang et al., 2013; Wu et al., 2016; Li et al., 2017; Xiao et al., 2017), but its development history, and  regional hydrology due to climate changes can still be further elaborated. For example, some records suggest that the precipitation in Hani decreased from the early to late Holocene (Hong et al., 2010; Hong et al., 2009, 2010) while some others suggest it increased gradually (Zheng et al., 2018). In addition, some studies show that the precipitation during the YD had increased in the Hani peatland (Zhou et al., 2010; Hong et al., 2010), while others do not show obvious increases (Li et al., 2017).

In this study, two sediment cores were collected in Hani peatland. The $^{14}C$ ages, organic contents and Rb/Sr ratios over the past 13.8 ka were obtained to discuss the climate changes in northeast China and the evolution of EASM.



## 2 Materials and methods

### 2.1 Geographic setting

Hani peat (42°13′27″ N, 126°30′42″ E) is situated in a valley of Liuhe county, Tonghua city, Jilin province of China (Fig.1). This swamp is west of Changbai Mountain and was originally formed by a volcanic barrier lake. It covers an area of about

18 km² (Hong et al., 2009), with a mean surface elevation of 900 m. This swamp is saturated all year round, a peat-forming environment has been sustained since the late Pleistocene, and peat has accumulated to a depth of more than 9 m (Zhou et al., 2010). It is the deepest and fastest growing modern swamp in northeast China.

This area has a continental humid climate with long cold winters and short cool summers. In any given year, the frosty season is about 250 days and freezing weather continues from November to April. The annual mean temperature is 4 ℃ with

the highest of 22 ℃ in July and the lowest of -18 ℃ in January. The annual mean precipitation is about 750 mm; it mostly falls from May to September, and originates from the EASM (Fig.2). The regional vegetation here is a mixed forest of temperate deciduous, broad-leaved and coniferous trees. (Hong et al., 2009; Zhou et al., 2010)

### 2.2 Sampling and analysis methods

Two sediment cores, H2 and H3, with cumulative depths of 6.90 m and 8.24 m respectively, were recovered with a peat drill

near the center of the peatland. The distance between the two boreholes was about 20 m. Because the subsurface layer was mainly composed of loose *Sphagnum* and plant residue, the 0 to 1 m of H2 returned no core sample and the top 1 m of H3 was compressed to only about 43 cm. The majority of sediment in H3 is peat, except for the sandy layer from 5.25 to 5.60 m (Fig.3).

H3 was separated into subsamples at 1 cm depth intervals. To obtain high resolution chronology data, 13 samples were

selected for AMS$^{14}$C dating. These were pretreated with 10 % HCl, 10 % NaOH and 2 % NaClO successively to remove carbonates and organic matter. Then the samples were washed with deionized water and sieved, extracting its sporopollen concentrates for AMS$^{14}$C dating. The sporopollen extraction and AMS$^{14}$C dating were completed at the Pollen Laboratory and Accelerator Mass Spectrometry Center, Institute of Earth Environment, Chinese Academy of Science (IEECAS), respectively. The AMS$^{14}$C age was calibrated to calendar age using the IntCal 13 curve in Calib 7.0 software. The age-depth

model was established based on the Bayesian age-depth modeling in Bacon.R program. The calibrated age was finally interpolated and extrapolated using the OriginPro 9.1 to derive the age of the whole sediment sequence.

In order to clarify the change of organic matter in Hani peat, 456 samples were chosen from H3 at intervals of 2 cm (1.08-2.58 m with 1cm interval) for LOI$_{550℃}$ and organic content measurements. These samples were dried at 50 ℃ and weighed to about 0.35 g, ground to a powder, and then placed in a muffle furnace for 4 hours at 550 ℃. This was conducted at the Lake

Sediment Laboratory of IEECAS in Xi'an.

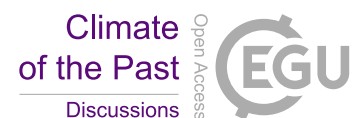

Before subsampling of H2, 1cm-resolution XRF scanning was first put into practice at the Core Scanning Laboratory of IEECAS to estimate the variation of elemental composition. After that, 1 cm-interval separation was carried out. In order to compare the consequence of H2 with H3, 4 cm-resolution $LOI_{550℃}$ analysis was also done on H2 samples using the same method as for H3.

## 3 Results

### 3.1 $^{14}$C chronology

The age-depth model (Fig.3) was established from the dating results of H3 (Table 1). It appears that the deposition rate of Hani peat been relatively stable, with an annual mean value of about 0.05 cm a$^{-1}$ and 0.09 cm a$^{-1}$ during the past 10.3 ka and 13.8-10.3 cal ka BP, respectively. The samples were collected in 2017, so the surface age was assumed to be -67 cal a BP. The basal age of 13.752 cal ka BP was obtained through extrapolation, thus the sequences of Hani peat recorded the climate change since about 13.8 cal ka BP ago.

### 3.2 Organic content and Rb/Sr ratio

The $LOI_{550℃}$ values represent the organic content changes in Hani peat. The depth-$LOI_{550℃}$ curves of H2 and H3 showed consistent variations (Fig. 4), also suggesting that the deposition of Hani peat was relatively stable. Therefore, the age of H3 was used in H2, and the age-$LOI_{550℃}$ curves of the two cores were established (Fig. 5e). The $LOI_{550℃}$ results showed that most of the values were higher than about 60 % except for the periods 11.3-10.3 cal ka BP and 2.0-1.4 cal ka BP. The abrupt decrease of $LOI_{550℃}$ during 11.3-10.3 cal ka BP corresponded to a sandy layer at 5.25-5.60 m depth which suggested that there was a changed deposition event at that time (Fig.5a, b, c). The $LOI_{550℃}$ values during 2.0-1.4 cal ka BP was slightly lower than 60 %, which may also reflect an anomalous sediment event similar to that during 11.3-10.3 cal ka BP. The Rb/Sr ratios of H2 produced from the XRF scanning data also show dramatic changes during the events around 11.3-10.3 cal ka BP and 2.0-1.4 cal ka BP (Fig.5d). The current evidence is insufficient to discuss the dynamic mechanisms of the two deposition events here, and will form part of a later work. In the present we focus on the proxies except the two periods 11.3-10.3 cal ka BP and 2.0-1.4 cal ka BP.

Excluding the effects of the two deposition events, $LOI_{550℃}$ changes of peat components (Fig. 6g) showed that the mean organic content of Hani peat was about 79 %, with the lowest value of about 58 % and the highest value of about 91 %. The organic content was lower during 13.8-11.4 cal ka BP. After the YD event, it increased gradually in the early Holocene (11.4-6.3 cal ka BP) and reached the highest value at around 6.3 cal ka BP. And then, the organic content continued to decline from the mid to late Holocene. Except for the long-term trend, there were also some abrupt declines in organic matter content, at 8.2 ka, 5.5 ka and 0.22 ka. Rb/Sr ratios were negatively correlated with $LOI_{550℃}$ values (Fig. 6f), it was relatively



high during 12.3-11.4 cal ka BP, and declined in the early Holocene, and reached its lowest value at about 6.7 cal ka BP, then increased gradually during the mid to late Holocene.

## 4 Discussion

### 4.1 Orbital scale climate changes recorded by Hani peat and its possible dynamics

Despite there being many studies on the orbital scale changes of the EASM using Hani peat parameters (Seki et al., 2009; Zhou et al., 2010; Zheng et al., 2011b, 2017, 2018; Hong et al., 2005, 2009; Hong et al., 2010; Li et al., 2017), no clear consensus has been reached. A part of those studies presented that the precipitation in Hani was decreased from early to late Holocene (Hong et al., 2009), while some others suggest that it increased across the entire Holocene (Zheng et al., 2018). In addition, some records show that the precipitation increased from early to mid Holocene time but then decreased from the mid to late Holocene (Li et al., 2017).

Here the $LOI_{550℃}$ and Rb/Sr ratio analyses were carried out to identify changes in organic matter content and the degree of chemical weathering. The change of organic contents in peat is directly related to vegetation development. In general, a warm/humid environment favours organic matter accumulation, when vegetation flourishes, initial productivity and organic debris input is high (Chai, 1990). Therefore, the $LOI_{550℃}$ values probably reflect the favourable temperature and precipitation conditions. The geochemical characteristics between Rb and Sr are obviously different. Rb has a strong affinity with clay while Sr readily enters into solution, so they are often fractionated (Goldstein and Jacobsen, 1988; Chen et al., 1999). With the enhancement of basin chemical weathering level, the Rb/Sr ratio of residual parts would increase. Correspondingly, Rb/Sr ratios in lake and peat sediments decrease (Jin et al., 2001). Therefore, warm/humid conditions usually correspond to low Rb/Sr ratios while cold/dry environments correspond to high Rb/Sr ratios. Rb/Sr ratios in well-preserved peat sediments actually indicate the leaching extent from source regions, which further reflects precipitation and temperature conditions in basins.

Based on the above information, the gradually increased $LOI_{550℃}$ values and decreased Rb/Sr ratios from early to mid Holocene in Hani peat (Fig. 6f, g) probably reflects an increase of temperature or precipitation or both. However, according to previous work, it is claimed that the temperature in Hani had no obvious uptrend from early to mid Holocene (Fig. 6a, b) (Hong et al., 2009; Zheng et al, 2018), thus the increased $LOI_{550℃}$ and decreased Rb/Sr ratios from early to mid Holocene were probably caused by an increase in precipitation (Fig. 6c, d, e) (Li et al., 2017; Zheng et al., 2018), reflecting that the EASM was enhanced from early to mid Holocene time. From the mid to late Holocene, the $LOI_{550℃}$ values decreased and the Rb/Sr ratios increased (Fig. 6f, g), which probably indicates a weakening of EASM intensity.

In addition, the changes of $LOI_{550℃}$ and Rb/Sr ratios are consistent with other EASM records from northern China monsoon frontal area, including the reconstructed precipitation of Gonghai (Fig. 7i) (Chen et al., 2015), tree pollen of Daihai (Fig. 7f) (Xiao et al., 2004) and Qinghai Lake (Fig. 7h) (Shen et al., 2005), reconstructed water level of Hexiazi Island (Fig. 7e)



(Zhang et al., 2018), as well as magnetic susceptibility (MS) of the Yulin loess-paleosol section (Fig. 7g) (Lu et al., 2013). These records all show that the EASM gradually strengthened from the early to mid Holocene, and reached its strongest intensity at mid Holocene time (about 6-7 cal ka BP), then weakened gradually.

The evolution of EASM during the Holocene was probably influenced by the coefficient of the solar insolation and northern

hemisphere ice volumes. From early to mid Holocene, the solar insolation decreased gradually (Fig. 7a) (Berger and Loutre, 1991), but the cold forcing from the northern hemisphere also declined (Haug et al., 2001) due to the melting of the ice sheets (Fig. 7b) (Dyke, 2004). The impacts of northern hemisphere ice volume changes seem to be more important than the solar insolation on the monsoon intensity in northern China, thus the EASM gradually increased in response to the reduced ice volume during early to mid Holocene (Lu et al., 2018; Berger and Loutre, 1991; Chen et al., 2015; Lambeck et al., 2014;

Carlson et al., 2008). By about 7 cal ka BP, the ice sheet had retreated to a position similar to today (Dyke, 2004; Törnqvist and Hijma, 2012) (Fig. 7b) (Dyke, 2004). We judge that the EASM weakened gradually from the mid to late Holocene, corresponding to a decline in solar insolation (Fig. 7a) (Berger and Loutre, 1991).

### 4.2 Millennial scale climate changes recorded by Hani peat

Many studies have focused on identifying abrupt climate change events in the EASM frontal area, such as the YD event

during 12.8-11.4 cal ka BP (Schettler et al., 2006; Zhou et al., 2010; Hong et al., 2010; Zheng et al., 2011a, b; Ma et al., 2012; Wu et al., 2016; Sun et al., 2016; Li et al., 2017). Some simulation studies have proposed that the freshwater input into the North Atlantic during the YD event caused a weakening of AMOC, resulting in a southward migration of ITCZ and decreased EASM intensity (Chiang et al., 2014; Renssen et al., 2015).

The climatic conditions in northeast China during the YD event have been part of many studies, including work based on

Hani peat (Hong et al., 2010; Zhou et al., 2010; Zheng et al., 2011a, b; Li et al., 2017). The decreased temperature in northeast China during the YD event has been demonstrated from some proxy records, but the cause in variation in precipitation still remains unclear. Some studies in Hani suggested that the climate was humid during YD (Hong et al., 2010; Zhou et al., 2010; Zheng et al., 2011b), but others argue for  no significant increase in precipitation (Li et al., 2017). In this study, an evident YD was found in the $LOI_{550℃}$ and Rb/Sr records of Hani peat. During the YD event, the organic content

was low and the chemical weathering rate was reduced, which were possibly influenced by lower temperature and/or decreased precipitation. Previous studies (Hong et al., 2010; Zheng et al., 2018) show a decline in temperature during the YD, which could alone result in the weakening of chemical weathering and the decrease of organic matter content. It is difficult to confirm if the reduction in both organic contents and chemical weathering degree could reflect just the decline of temperature or both temperature and precipitation. Therefore, it is still hard to draw a definite conclusion about the change of

precipitation in northeast China during the YD on current evidence. Moreover, it is remarkable that the climate within the YD (12.8-11.4 cal ka BP) was also unstable, with some dramatic fluctuations in organic content and Rb/Sr ratio profiles.



Following the YD, there were also some millennial scale rapid changes in $LOI_{550°C}$ and Rb/Sr records of Hani peat during the Holocene, such as 8.2 ka, 5.5 ka and LIA (Fig. 7). The 0.5-2 ka filtering analysis was done for $LOI_{550°C}$ data of H3, and the result shows that abrupt changes of the Hani peat organic content record was correlated with the drift ice indices of the North Atlantic and the atmospheric $\Delta^{14}C$ record (Fig. 8), indicating that both North Atlantic climate change and solar activity
probably influenced the Holocene millennial scale abrupt changes of the EASM (Bond et al., 2001).

## 5 Conclusions

In this study, two sediment cores were collected from Hani peat of northeast China and AMS$^{14}C$ dating shows that these cores covered the past 13.8 ka. $LOI_{550°C}$ and XRF scanning analysis were carried out to obtain the organic matter contents in peat sediment and the chemical weathering degree in basin, both of which could be used to reflect the climate variations in
Hani peatland. The results suggested that the precipitation at Hani probably increased gradually from 11.4 to 6.3 cal ka BP and reached the wettest conditions at about 6.3 cal ka BP, possibly linked to an enhanced EASM from early to mid Holocene. From 6.3 cal ka BP to present, both the organic matter content and the chemical weathering degree gradually decreased, probably indicating that the EASM declined continuously during the mid to late Holocene. These and other published EASM records from northern China, demonstrate that the evolution of EASM during the Holocene was likely to be dominated by
the combination of the influences from changing solar insolation and the northern hemisphere ice volumes. In addition, there were also some millennial scale abrupt climate events evident in the records of Hani peatland, including the YD, 8.2 ka, 5.5 ka and LIA. The teleconnections between abrupt changes of Hani peat and the drift ice indices of North Atlantic as well as the atmospheric $\Delta^{14}C$ record indicated that both North Atlantic climate changes and solar activity probably had significant impacts on the millennial-scale EASM variations.

**Author contribution**

Ge Shi compeleted the main experiments and data analysis. Hong Yan provided the major founding support and theoretical direction. Wenchao Zhang and Haobai Fei did the sampling work and assisted in indoor subsampling. Shuanshuan Cao and Fengyan Lu preprocessed the dating samples. Chengcheng Liu and Xiaolin Ma assisted in consulting the literature. Weijian Zhou provided the platform for dating. John Dodson, Henk Henjins and Zhisheng An provided
theoretical supports. Finally, Ge Shi prepared the manuscript with contributions from all co-authors.

**Acknowledgements**

Financial support for this research was provided by the National Natural Science Foundation of China (NSFC) (41877399, 41522305) and the research Projects from Chinese Academy of Sciences (QYZDB-SSW-DQC001 and





132B61KYSB20160003) and Qingdao National Laboratory for Marine Science and Technology of China (QNLM2016ORP0202). This study is a part of the "Belt & Road" project of IEECAS.

**Competing interests**

The authors declare that they have no conflict of interest.

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



**Table**

**Table 1 Dating results of Hani peat core H3. The pollen concentrates were used for AMS$^{14}$C dating and the dating results were calibrated by the IntCal 13 curve in Calib 7.0 software.**

| Sample no. | Depth (cm) | Material | $^{14}$C age (a BP) | Error (1σ) | Median cal age (cal a BP) |
|---|---|---|---|---|---|
| H70 | 127 | pollen | 1690 | 25 | 1591 |
| H170 | 227 | pollen | 3420 | 25 | 3667 |
| H270 | 327 | pollen | 5020 | 30 | 5771 |
| H380 | 437 | pollen | 7265 | 35 | 8092 |
| H513 | 501 | pollen | 8860 | 30 | 10016 |
| H533 | 521 | pollen | 9210 | 35 | 10363 |
| H553 | 541 | pollen | 9310 | 30 | 10523 |
| H560 | 548 | pollen | 9320 | 35 | 10532 |
| H585 | 573 | pollen | 9590 | 40 | 10932 |
| H601 | 589 | pollen | 9920 | 30 | 11302 |
| H670 | 658 | pollen | 10330 | 40 | 12159 |
| H770 | 758 | pollen | 11040 | 40 | 12905 |
| H828 | 816 | pollen | 11820 | 50 | 13650 |

**Figure captions**

**Fig.1** Location of Hani peatland and research sites mentioned in this study. The black star indicates the location of Hani peatland, the solid dots represent the Hexiazi Island (Zhang et al., 2018), Gushantun (Zheng et al., 2018), Daihai Lake (Xiao

10 et al., 2004), Gonghai Lake (Chen et al., 2015), Qinghai Lake (Shen et al., 2005) and Yulin loess (Lu et al., 2013), respectively. The dashed black arrow leads to the sampling site of this study.

**Fig.2** Monthly mean precipitation and temperature of Jingyu station near the Hani peat from 1981 to 2010 (Li et al., 2017).

15 **Fig.3** The age-depth model of Hani peat H3 core. Left is the lithological column of Hani peat H3 core. Right is the age-depth model that established using the Bayesian age-depth modeling in Bacon.R program.



**Fig.4** The LOI$_{550℃}$ profiles of the two cores in Hani peat based on depth. The blue solid line represents the LOI$_{550℃}$ profile of core H2 and the red solid line represents the LOI$_{550℃}$ profile of core H3. The grey dotted lines indicate the depth of AMS dating samples and the numbers in parentheses are the corrected dating ages.

**Fig.5** Two abnormal sedimentary events of Hani peat. (d) The Rb/Sr ratio profile of H2; (e) The LOI$_{550℃}$ profiles of H2 and H3. The two grey bars indicate two abnormal sedimentary events at 11-10 cal ka BP and 1.8-1.5 cal ka BP. During these two events, the Rb/Sr ratios and the LOI$_{550℃}$ values both decreased abruptly. The sediment can be observed clearly changing from peat (a, c) to sand (b) during 11-10ka.

**Fig.6** Comparison of Hani peat proxy records of this study with other paleoclimate records from northeast China. (a) $\delta^{18}O$ record of Hani peat cellulose, which was used as a temperature sensitive proxy (Hong et al., 2009); (b) mean annual air temperature (MAAT$_{peat}$) reconstruction from Hani peat (Zheng et al., 2018); (c) the percentage of sediments with grain size between 37.0-497.8μm in Hani peat, which was used to reflect the intensity of EASM precipitation (Li et al., 2017); (d) pH in Gushantun peat, the higher values was used to indicate lower precipitation (Zheng et al., 2018); (e) pH in Hani peat
(Zheng et al., 2018); (f) Rb/Sr profile of Hani peat in this study; (g) LOI$_{550℃}$ profile of Hani peat in this study.

    **Fig.7** Comparison of Hani peat proxy records of this study with other EASM reconstructions from northern China's monsoon frontal area. (a) 60 °N insolation (Berger and Loutre, 1991); (b) Northern Hemisphere ice sheet extent (Dyke, 2004); (c) Rb/Sr ratios of Hani peat in this study; (d) LOI$_{550℃}$ values of Hani peat in this study; (e) the quantitative reconstruction of
water level through integrated curve of magnetic susceptibility, carbonate content, Md and sand fraction in Hexiazi Island, the higher water level reflects a stronger EASM intensity (Zhang et al., 2018); (f) tree pollen of Dahai Lake (Xiao et al., 2004); (g) Magnetic Susceptibility (MS) of Yulin loess-paleosol section (Lu et al., 2013); (h) tree pollen of Qinghai Lake (Shen et al., 2005); (i) reconstructed annual mean precipitation (P$_{ANN}$) based on pollen contents in Gonghai Lake (Chen et al., 2015).


    **Fig.8** Comparison of the 0.5-2ka filtering results of Hani LOI$_{550℃}$ data with the Bond Cycle indexes and solar irradiance record. (a) Smoothed and detrended $^{14}C$ production rate record. The high values reflect the decreases of solar irradiance (Bond et al., 2001); (b) Hematite- stained grains of MC52-VM29-191, expressed as percentages of lithic grains (ice-rafted debris), the grey numbers above represent eight drift ice events in North Atlantic during the Holocene (Bond et al., 2001); (c)
0.5-2 ka filtering results of Hani LOI$_{550℃}$ data, the lower values correspond to the weaker solar irradiance and the higher ice-rafted debris contents. The data during the two abnormal sedimentary events was removed when analyzed.



**Figures**

**Fig.1**

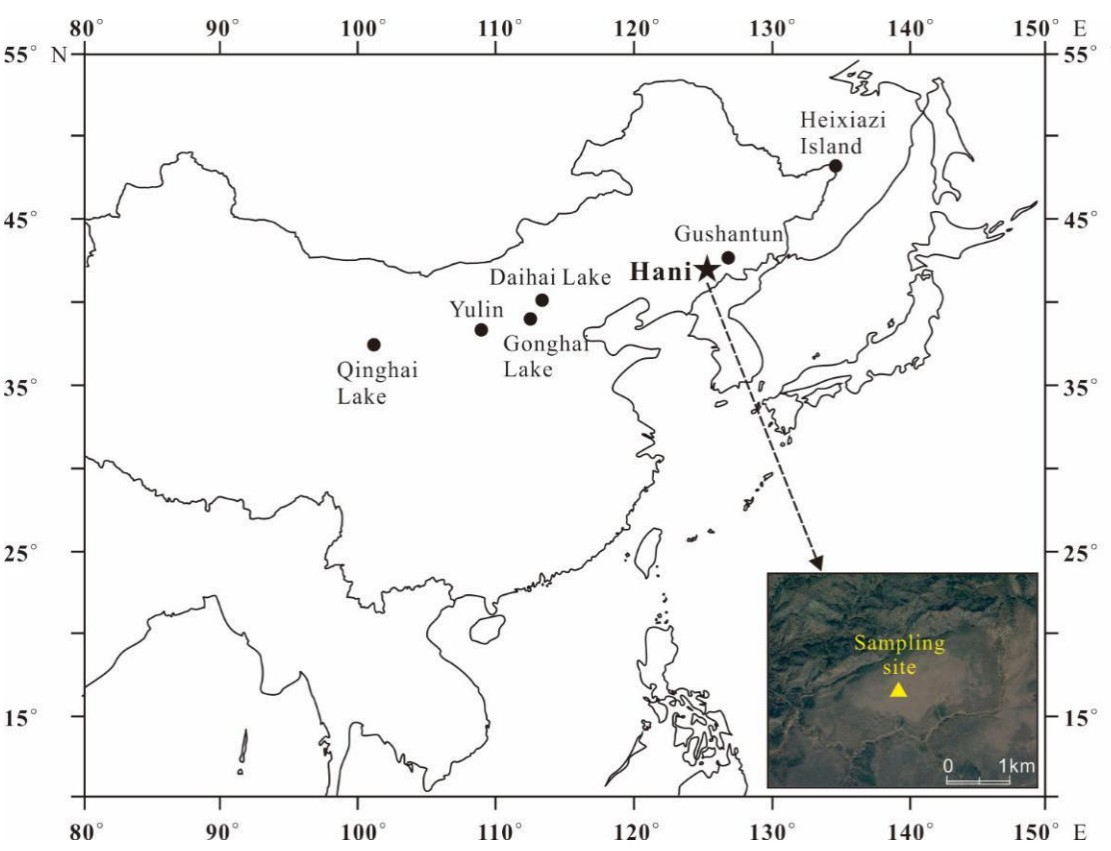



**Fig.2**

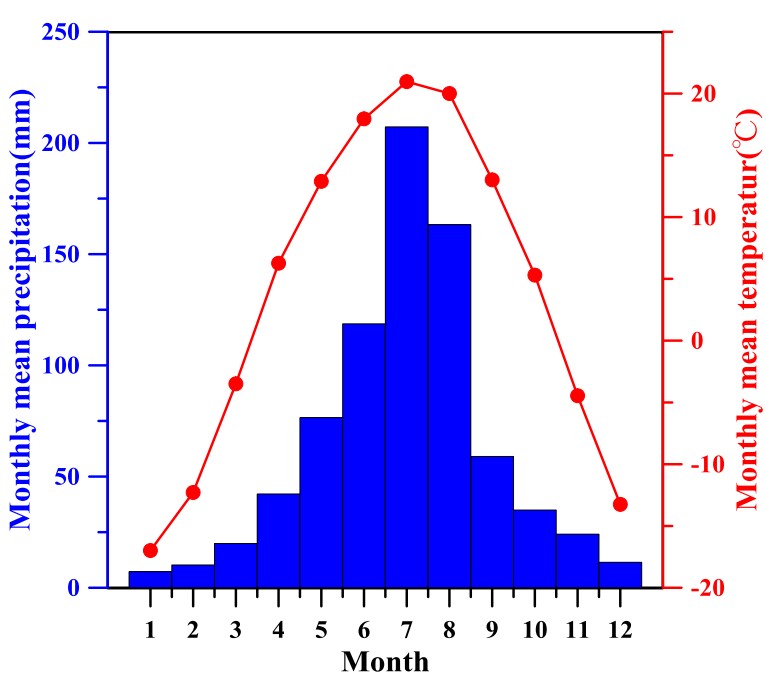

**Fig.3**

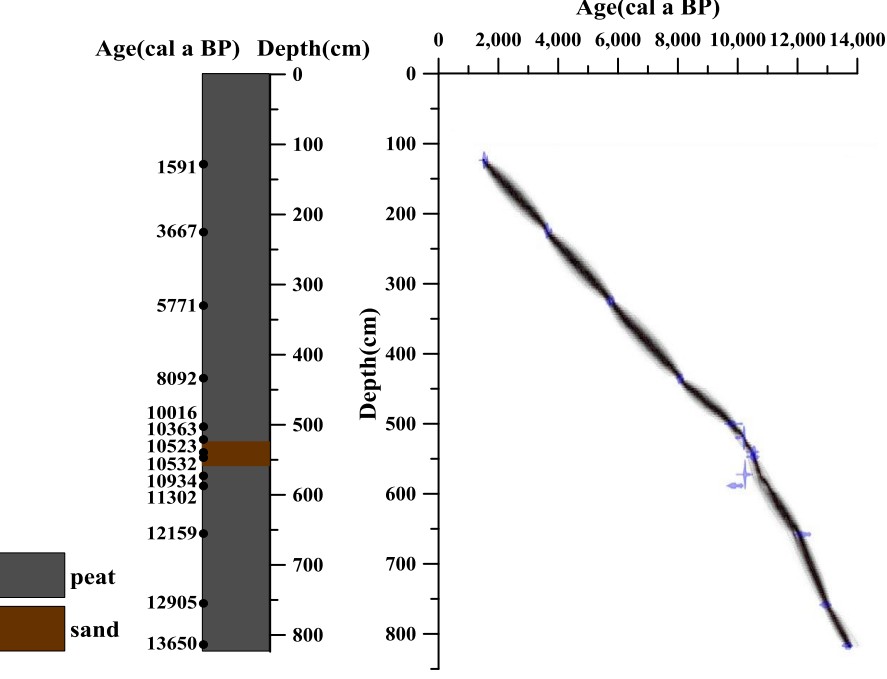

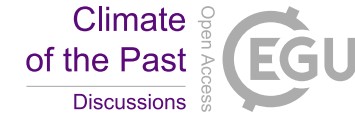

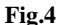

**Fig.4**

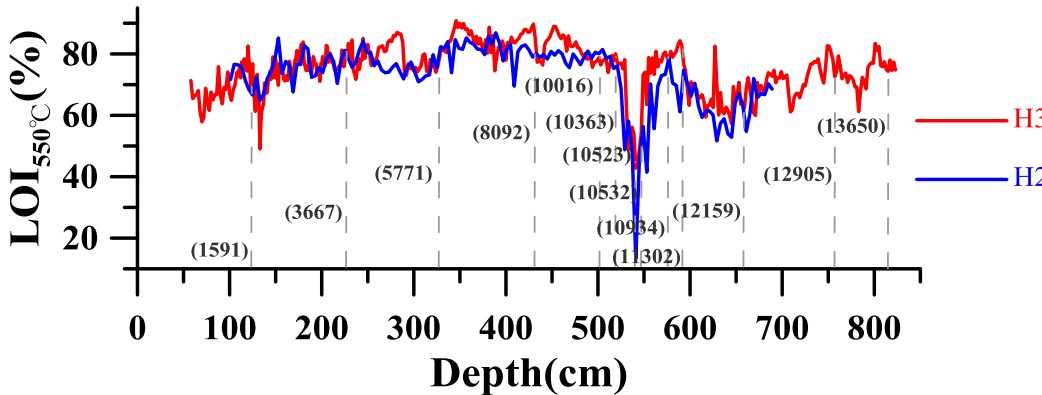

**Fig.5**

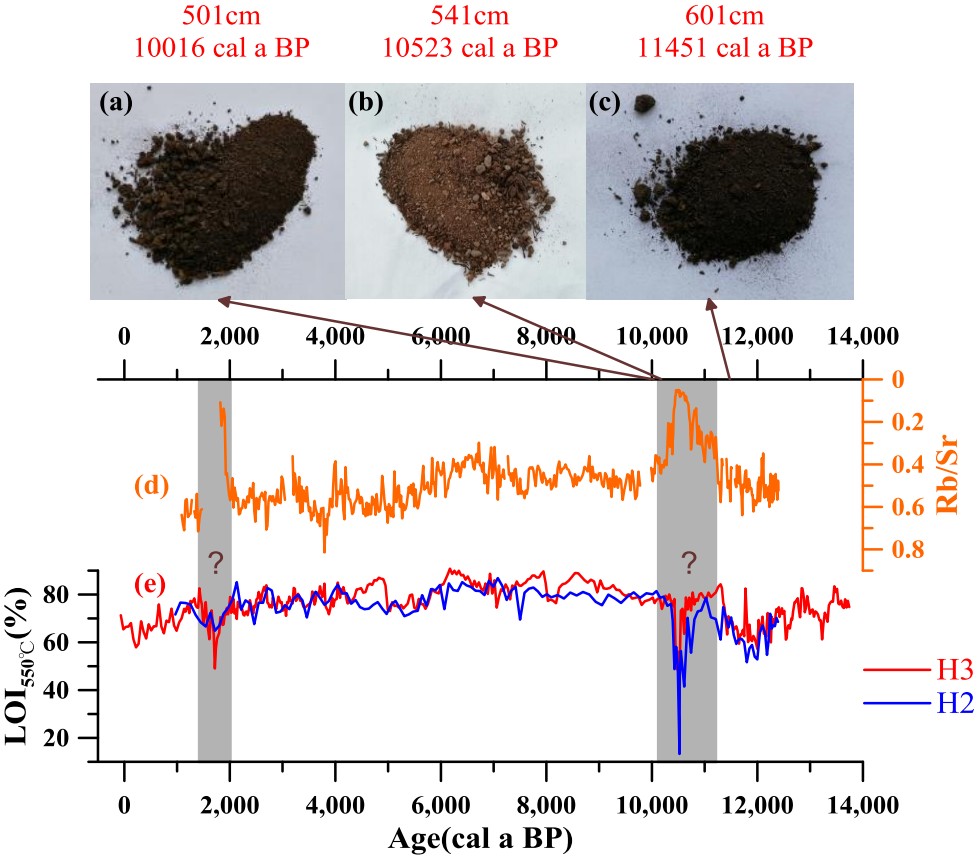


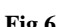


**Fig.6**

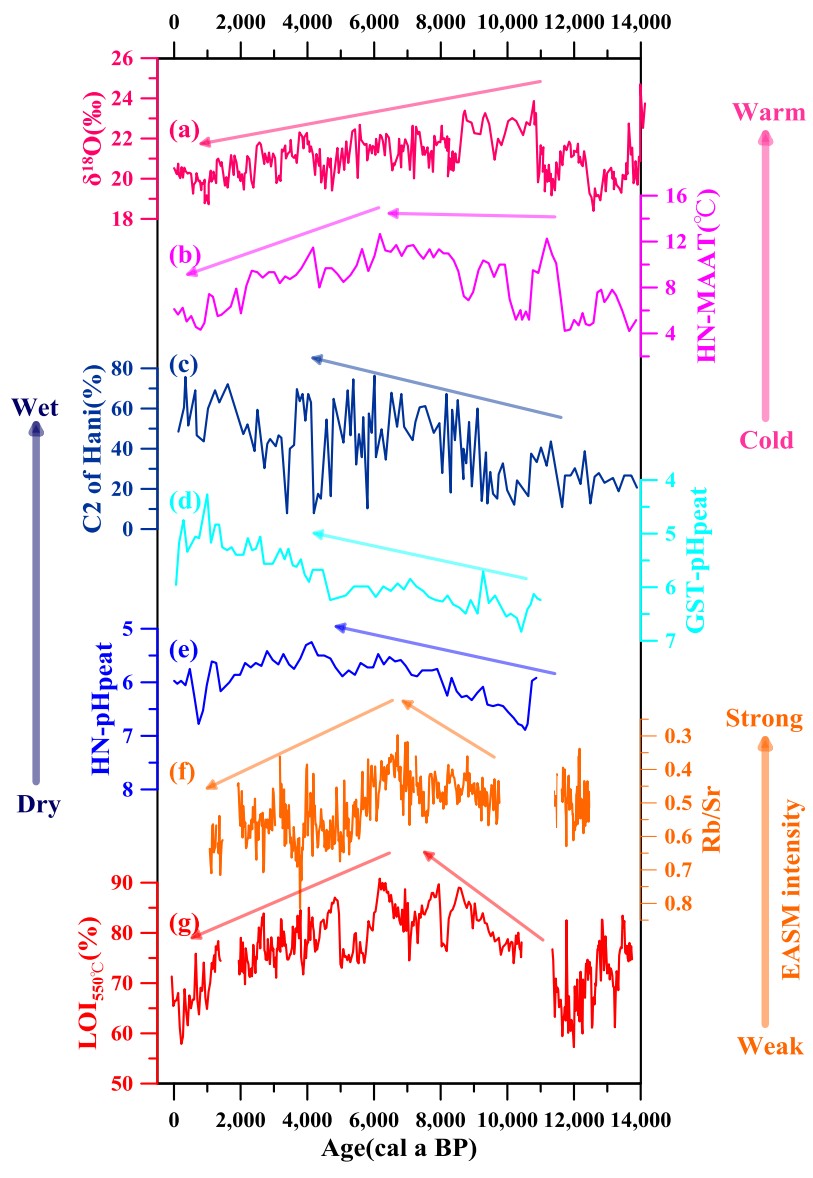





**Fig.7**

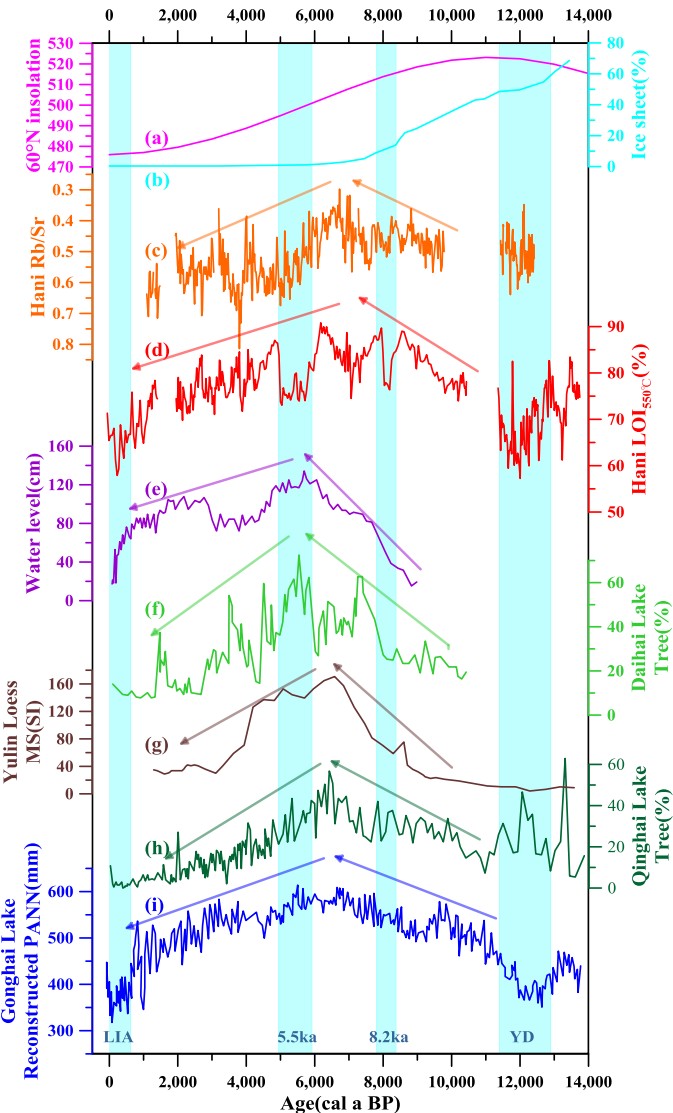

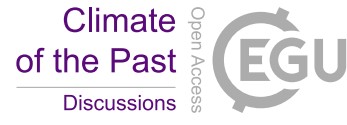

**Fig.8**

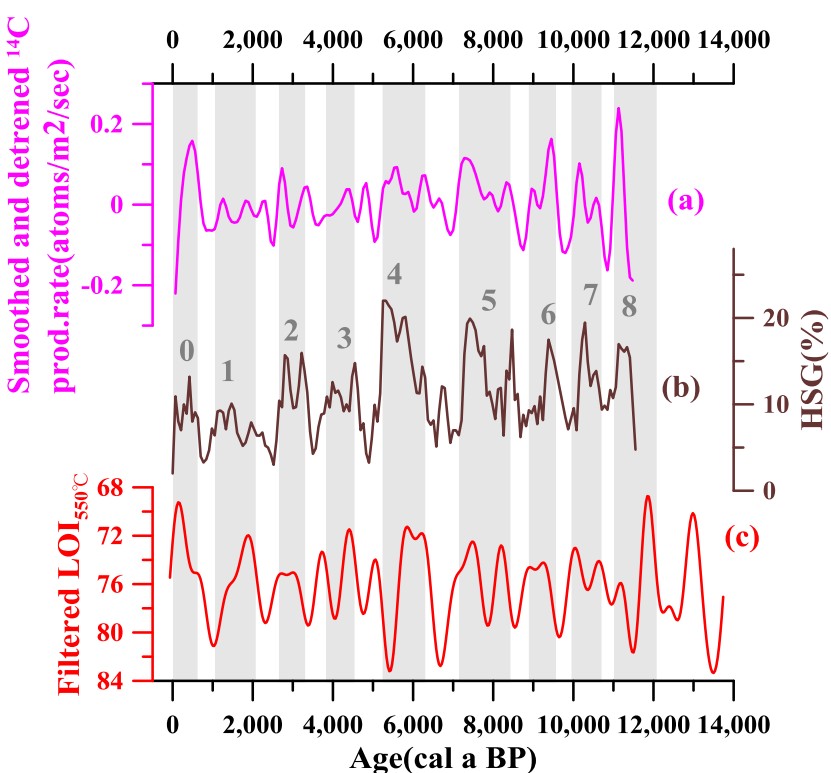