# Peer review of "Climate changes recorded by Hani Peat in Northeast China over the past 13.8 cal ka BP"

_Climate of the Past, 2019_

## Referee Comment (RC1) · Anonymous Referee #1 · 21 Apr 2019

Two records were collected in Hani peat-land. The LOI550°C (organic matter content) and Rb/Sr ratios (chemical weathering) over the past 13.8 ka were analysed to discuss the climate change in Northeast China and the evolution of East Asian Summer Monsoon (EASM). The organic matter content and chemical weathering were compared with previous lake and peat-land temperature and precipitation proxy records from northeast China. The manuscript is well presented and provides results of two new proxies research for northeast China that will contribute to the knowledge of the evolution of East Asian Summer Monsoon (EASM) and adds relevant information to improve our understanding of the past climate based on multi-proxy climatic records in this region. The text mentions a long-term "positive correlations" trend between LOI 550°C and Rb/Sr ratio over the past 13.8 ka (decrease Rb/Sr ratio value while increase

LOI 550°C value) (Figure 6f,g). Also, three specific events marked by an abrupt decline in organic matter content at c. 8.2ka, 5.5 ka and 0.22 ka are identified. These events are "negatively correlated" with Rb/Sr ratio value (increase Rb/Sr ratio value while LOI 550°C value decrease) (Figure 7c,d). However, I am concerned about the choice of time periods, when the authors used the terms "positive correlations" and negatively correlated" (see below) and, about the figure 6t does not show Rb/Sr ratio value for the last 1.0 ka, thus it is not possible to see any correlation between LOI 550°C and Rb/Sr ratio by 0.22ka (as the authors present). The figure 6f and g do not show clear decreases in Rb/Sr ratios from early to mid Holocene in the Hani peat record (c.12-7.0 ka.). The strength of the interpretations depend on where and when the starting point of the indicator arrow is chosen. For instance, if the horizontal line is drawn at 0.5, when the values for each element are the same (Figure 6f), the general trend of Rb/Sr ratio between (c.10-7.0 ka.) is stable, with values remaining around 0.5 (slightly strong EASM intensity during the mid-Holocene). Then, the lowest Rb/Sr ratio value between c.7.0-6.0 ka suggests a strong EASM intensity, followed by general trend to increase in Rb/Sr ratio value (less strong/weak EASM intensity) towards present. The same comments for LOI 550°C, if the horizontal line is drawn in Figure 6g at 80% of LOI 550°C (water content versus organic content is linear up to loss ignition =80%.), the figure show a general warmer climatic conditions between c.9.0-5.0 ka suggests a strong EASM intensity (suc. mag) during the mid-Holocene, followed by general trend to increase in Rb/Sr ratio value (less strong/weak EASM intensity) towards present. In addition, this work highlights the decrease in LOI 550°C and the correlation with the Rb/Sr ratio value, thus it is key to show any tephra layers in the stratigraphy to exclude them for the interpretation. The authors do not discuss two major periods of abrupt change shown in their work, these events show the lowest organic matter content due to the sandy layer deposition and the lowest Rb/Sr ratio by c. 11.3-10.3 cal ka BP and c. 2.0-1.4 cal ka BP, respectively. The authors suggest that there insufficient evidence to discuss "the dynamic mechanisms of the two depositions events". However, there are previous works about tephra deposition in Hani peat-land coeval to both deposition
events (Huang et al., 2015; Zhao et al., 2016 among others). The drastic decrease in LOI 550°C by c. 8.2ka and the increasing of Rb/Sr ratio suggests a cold-dry climatic event (reduction of weathering intensity), consistent with falling in temperature is slight decrease in precipitation (Figure 6acdg) in other published Hani peat proxy records. The evidence suggests a weak East Asian monsoon (hot/wet summer) during the 8.2 ka. event highlighting the sensitivity of the peat for EASM reconstruction. The comparison between Rb/Sr ratio and CaCO2 (previous works) would have allowed stronger interpretations of specific time intervals Hani peat, because the Rb/Sr ratio appear to be a response to moisture conditions or effective moisture conditions (figure 7f). An increase of Rb/Sr ratio is consistent with increase in percentage tree cover at Daihal Lake, except during the short-time c.8.2ka event, perhaps because of the low resolution pollen intervals around 8.2ka.

---

## Short Comment (SC1) · 5 May 2019

Question 1: Two records were collected in Hani peat-land. The LOI550°C (organic matter content) and Rb/Sr ratios (chemical weathering) over the past 13.8ka were analyzed to discuss the climate change in Northeast China and the evolution of East Asian Summer Monsoon (EASM). The organic matter content and chemical weathering were compared with previous lake and peat-land temperature and precipitation proxy records from northeast China. The manuscript is well presented and provides results of two new proxies research for northeast China that will contribute to the knowledge of the evolution of East Asian Summer Monsoon (EASM) and adds relevant information to improve our understanding of the past climate based on multi-proxy climatic records in this region. The text mentions a long-term "positive correlations" trend between

LOI550°C and Rb/Sr ratio over the past 13.8ka (decrease Rb/Sr ratio value while increase LOI550°C value) (Figure 6f, g). Also, three specific events marked by an abrupt decline in organic matter content at c. 8.2ka, 5.5ka and 0.22ka are identified. These events are "negatively correlated" with Rb/Sr ratio value (increase Rb/Sr ratio value while LOI550°C value decrease) (Figure 7c, d). However, I am concerned about the choice of time periods, when the authors used the terms "positive correlations" and negatively correlated" (see below) and, about the figure 6t does not show Rb/Sr ratio value for the last 1.0ka, thus it is not possible to see any correlation between LOI550°C and Rb/Sr ratio by 0.22ka (as the authors present).

Answer: Thank you for the comment. The LOI550°C and Rb/Sr ratio data were obtained from core H3 and H2, respectively. The top 1m sediments of H2 was lost during the sample collection, and thus we do not have the Rb/Sr ratio data for the last 1ka to compare with LOI550°C data. The "positive correlations" and "negatively correlated" in the manuscript were used to describe the long-term trend of our data and your comments make us realize that this statement may not be appropriate. We changed these ambiguous description in the revision.

Question 2: The figure 6f and g do not show clear decreases in Rb/Sr ratios from early to mid Holocene in the Hani peat record (c.12-7.0ka.). The strength of the interpretations depend on where and when the starting point of the indicator arrow is chosen. For instance, if the horizontal line is drawn at 0.5, when the values for each element are the same (Figure 6f), the general trend of Rb/Sr ratio between (c.10-7.0ka.) is stable, with values remaining around 0.5 (slightly strong EASM intensity during the mid-Holocene). Then, the lowest Rb/Sr ratio value between c.7.0-6.0 ka suggests a strong EASM intensity, followed by general trend to increase in Rb/Sr ratio value (less strong/weak EASM intensity) towards present. The same comments for LOI550°C, if the horizontal line is drawn in Figure 6g at 80% of LOI550°C (water content versus organic content is linear up to loss ignition =80%.), the figure show a general warmer climatic conditions between c.9.0-5.0ka suggests a strong EASM intensity (suc. mag)

during the mid-Holocene, followed by general trend to increase in Rb/Sr ratio value (less strong/weak EASM intensity) towards present.

Answer: Thanks. The time interval of YD event in our records is about from 12.5ka to 11.4ka. In addition, the data between 11-10 ka were disturbed by rapid deposition event caused by volcanoes or floods. Thus, we can only discuss the Holocene EASM variations since the 10ka. The Rb/Sr ratios decreased slightly from 10ka to 6.7ka and increased since the 6.7ka. Meanwhile, the LOI550°C values increased gradually from 10-6.3ka and decreased since the 6.3ka. The consistent changes of two records indicated that the EASM strengthened gradually in the early Holocene, weakened in the late Holocene, and the strongest in the middle Holocene.

Question 3: In addition, this work highlights the decrease in LOI550°C and the correlation with the Rb/Sr ratio value, thus it is key to show any tephra layers in the stratigraphy to exclude them for the interpretation. The authors do not discuss two major periods of abrupt change shown in their work, these events show the lowest organic matter content due to the sandy layer deposition and the lowest Rb/Sr ratio by c. 11.3-10.3 cal ka BP and c. 2.0-1.4 cal ka BP, respectively. The authors suggest that there insufficient evidence to discuss "the dynamic mechanisms of the two depositions events". However, there are previous works about tephra deposition in Hani peat-land coeval to both deposition events (Huang et al., 2015; Zhao et al., 2016 among others).

Answer: Thank you for your advice. We thought at first that these two events should have been caused by volcanoes. In order to verify this hypothesis, we selected three samples at the depth of 128, 133, and 139cm to do scanning electron microscope analysis, but we didn't find efficient evident. In this case, we didn't discuss these two events and just focused on the EASM variations in this paper. We are designing further sampling plans and will do more to verify whether these two events were caused by volcanoes, floods or something else, then discussing specially in our later works.

Question 4: The drastic decrease in LOI550°C by c. 8.2ka and the increasing of Rb/Sr

ratio suggests a cold-dry climatic event (reduction of weathering intensity), consistent with falling in temperature is slight decrease in precipitation (Figure 6acdg) in other published Hani peat proxy records. The evidence suggests a weak East Asian monsoon (hot/wet summer) during the 8.2ka event highlighting the sensitivity of the peat for EASM reconstruction. The comparison between Rb/Sr ratio and CaCO3 (previous works) would have allowed stronger interpretations of specific time intervals Hani peat, because the Rb/Sr ratio appear to be a response to moisture conditions or effective moisture conditions (figure 7f). An increase of Rb/Sr ratio is consistent with increase in percentage tree cover at Daihai Lake, except during the short-time c.8.2ka event, perhaps because of the low resolution pollen intervals around 8.2ka.

Answer: Thanks for your comment. The higher Rb/Sr ratio in this paper represented a weaker chemical weathering degree, which indicated a colder/drier climate. The increased Rb/Sr ratio in our study is generally correspond to decreased tree pollen percentage at Daihai Lake, suggesting that the two records indicated similar EASM variations during the Holocene. At around 8.2ka, our Rb/Sr ratios increased while the LOI550°C decreased, indicating a cold/dry climate. However, the tree pollen record from Daihai Lake did not present significant changes during the 8.2ka, probably due to the relative low resolution of pollen record or regional difference. In addition, we also determined the LOI950°C to obtain the carbonate contents in Hani peat and the results showed that the lower carbonate contents corresponded to stronger chemical weathering degree, which further validated the EASM variations we obtained from Rb/Sr and LOI550°C records.

---

## Referee Comment (RC2) · Anonymous Referee #2 · 21 May 2019

The authors use two peat cores collected 20 m apart from Hani peatland in Northeast China to reconstruct climate, in particular East Asian summer monsoon variations, during the last 13,000 years. One of the cores was dated by AMS 14C dates on concentrated pollen grains. The main proxies they use include LOI (OM%), interpreted as reflecting vegetation productivity, and Rb/Sr, interpreted as reflecting chemical weathering. These two proxies were analyzed on these two separate cores, while the chronology for the second core was based on correlation of LOI results from both cores. They conclude that the summer monsoon increased from the early to mid-Holocene and then decreased from the mid- to late Holocene. Then they interpret these changes were due to the combined influences from insolation and ice volume.

I have several major concerns about the manuscript.

The proxy interpretation as presented in the manuscript is too simplistic and lacks support from peatland/peat accumulation process. I don't think that LOI/OM alone can be directly used as climate proxy without constraints from other data and information. Peatland vegetation and plants should produce peat with near 100% in-site OM (except perhaps minor components of biogenic silicates, such as from phytoliths in some plant tissues), almost entirely independent of peatland plant species composition or vegetation productivity. On the other hand, the mineral/inorganic component (= 100% - OM%) is mostly derived from outside of the peatland from the surrounding landscapes, which may potentially reflect regional climate. However, the mineral materials could be transported either by fluvial process, such as streams and overland flows, or by eolian input by winds. For example, an increase in mineral content (that is, decrease in OM%) could be caused by flooding (that is, wet environment and high precipitation), or by wind-blown dust input (that is, in a dry environment and low precipitation, due to sparse vegetation and mobilized/exposed top soils). Therefore, distinguishing these two opposite causes, often with additional independent proxies (such as pollen/vegetation), is essential for meaningful climate interpretations.

Rb/Sr ratios pretty much reflect the same process as mineral content (100% - LOI/OM%), but their difference could reflect the sensitivity of these two proxies to erosion (physical weathering) and chemical weathering on the surrounding watershed.

The authors explicitly indicate that they would not discuss the two mineral deposition events at 11.3-10.3 ka and 2.0-1.4 ka. Actually these two intervals should be interpreted as the same way as other intervals with low OM contents, likely caused by either fluvial or eolian process, but at large magnitudes. The authors should consider these two periods along with other fluctuations to generate a consistent interpretation. In any case, I don't think the proxies they use are adequate and robust enough to make convincing climate reconstructions.

Despite that the authors identify existing problems and open questions about the Asian summer monsoon by citing many references, I don't think that the record presented

here contribute much to the debate, due to the shortcoming of the proxy they use (see comments above). Also, they have to rely on previous work to distinguish temperature from precipitation changes, such as on page 5 lines 22-28, which also limit the new contribution from this study.

As such, I don't think that the large-scale climate discussions about insolation and solar forcing (Figs. 7 and 8) are supported by the evidence and arguments.

The number of references is excessive. The authors cite >5 pages of references for a 7-page manuscript! Most references are not needed.

In summary, based on the above considerations, I do not recommend the publication of the manuscript without additional analyses and improved proxy interpretations.

Specific comments: Page 1 Line 26: insolation = incoming solar radiation, so "solar insolation" is redundant.

Line 26: "ka" often refers to 1000 year BP. Here it is better to say "kyr".

Page 2 Line 9: delete "." after "China"

Line 17: I don't think Gorham (1991) is an appropriate reference for "studying past climate changes", as this is a seminal paper on peatland carbon stocks and their sensitivity to climate change, but not proxy/paleoclimate studies.

Line 18-20: this excessive citation is not needed.

Line 24-26: again too many references, which are unnecessary.

Line 29: I don't think "YD" has been defined (Younger Dryas). Define abbreviation when first used.

Page 3 Line 3: Hani peatland, not Hani peat.

Line 4: probably use "peatland", rather than "swamp", as they are different. Swamp is too specific for woody peat-accumulating system with fluctuating water table.

Line12: I'm not sure that these references are suitable for vegetation types, even though they may mention vegetation types, but likely were based on other sources. Also, move "." to the end of the sentence.

Line 21-22: why bothering with dating pollen grains? The peat contains high OM, and you should either find identifiable macrofossil or bulk peat for dating. If worrying about potential organic material from aquatic plants (usually not abundant on that type of peatlands), then pollen grains could come from aquatic plants as well (unless you pick specific types of pollen grains, such as large pine or spruce pollen grains). I don't necessarily question the results, but it just appears to me the effort is unnecessary.

Line 25-26: I think Bacon program would generate an age model, assigning age to every depth, so I'm not sure you need a plotting program (OriginPro) for interpolation.

Line 29: for LOI analysis, I'm not sure that weighted and grounded samples are necessary, unless grounded dry samples were also used for other analysis.

Page 4 Line 8. Delete "annual"

Line 21-22: These two mineral layers should be discussed in order to understand other changes in OM% (that is, mineral content). See my general comment above.

Line 12-15: this simplistic interpretation of LOI/OM is incorrect. No matter how productive is the peatland, if no external input of mineral materials, there will be no change in OM%. See my general comments above.

Line 18-21: This interpretation doesn't consider the transport process. Humid condition may cause great transport of fluvial-derived mineral materials, while dry condition may cause great source and transport of wind-blown mineral/dust. See my general comments above.

Page 7 Line 21: "major funding support"

Line 22: change "indoor" to "lab"?

Page 8 Too many references, and some of them are probably not necessary.

Page 14 Table 1. -indicate the unique dating lab ID number for each date, not just sample number. -indicate the depth range, such as 127-128 cm, or 126-127 cm?

Figure 1 Detail map of study site is essential to understand the potential sources of mineral materials, through fluvial or eolian inputs.

---

## Author Comment (AC2) · 18 Jun 2019

The authors use two peat cores collected 20 m apart from Hani peatland in Northeast China to reconstruct climate, in particular East Asian summer monsoon variations, during the last 13,000 years. One of the cores was dated by AMS 14C dates on concentrated pollen grains. The main proxies they use include LOI (OM%), interpreted as reflecting vegetation productivity, and Rb/Sr, interpreted as reflecting chemical weathering. These two proxies were analyzed on these two separate cores, while the chronology for the second core was based on correlation of LOI results from both cores. They conclude that the summer monsoon increased from the early to mid-Holocene and then decreased from the mid- to late Holocene. Then they interpret these changes were due to the combined influences from insolation and ice volume.

I have several major concerns about the manuscript.

1. The proxy interpretation as presented in the manuscript is too simplistic and lacks support from peatland/peat accumulation process. I don't think that LOI/OM alone can be directly used as climate proxy without constraints from other data and information. Peatland vegetation and plants should produce peat with near 100% in-site OM (except perhaps minor components of biogenic silicates, such as from phytoliths in some plant tissues), almost entirely independent of peatland plant species composition or vegetation productivity. On the other hand, the mineral/inorganic component (= 100% - OM%) is mostly derived from outside of the peatland from the surrounding landscapes, which may potentially reflect regional climate. However, the mineral materials could be transported either by fluvial process, such as streams and overland flows, or by eolian input by winds. For example, an increase in mineral content (that is, decrease in OM%) could be caused by flooding (that is, wet environment and high precipitation), or by wind-blown dust input (that is, in a dry environment and low precipitation, due to sparse vegetation and mobilized/exposed top soils). Therefore, distinguishing these two opposite causes, often with additional independent proxies (such as pollen/vegetation), is essential for meaningful climate interpretations.

Answer: Thank you very much for the helpful comment. As you said, there might be different interpretations about the OM% in peat sediments. On the one hand, the mineral content could be brought by wind, which caused the decreased OM% and reflect a dry environment condition. On the other, the mineral content could also increase due to the fluvial, which indicates a wet environment inversely. Therefore, it exactly would be much better to discuss it with independent proxies. Cui et al., 2006 and Yu et al., 2008 (in Chinese) ever did the paleo-vegetation study of Hani peatland. We compare our OM% record with the pollen records they provided and found that the two records show consistent variations during the Holocene (please see fig.1). This result supports our view that the higher OM% possibly reflects a warm/humid climate. We'll add the comparison result to our revision.

CPD
Reference:

1.Cui, M., Luo, Y., Sun, X. Paleoenvironmental and paleoclimate changes in Hani Lake, Jilin since 5ka BP. Marine Geology &Quaternary Geology, 26 (5): 117-122, 2006 (in Chinese).

2.Yu, C., Luo, Y., Sun, X. A high-resolution pollen records from Ha'ni Lake, Jilin, Northeast China showing climate changes between 13.1 ka cal. BP and 4.5 ka cal. BP. Quaternary Sciences, 28 (5): 929-938, 2008 (in Chinese).

2.Rb/Sr ratios pretty much reflect the same process as mineral content (100% - LOI/OM%), but their difference could reflect the sensitivity of these two proxies to erosion (physical weathering) and chemical weathering on the surrounding watershed. The authors explicitly indicate that they would not discuss the two mineral deposition events at 11.3-10.3 ka and 2.0-1.4 ka. Actually these two intervals should be interpreted as the same way as other intervals with low OM contents, likely caused by either fluvial or eolian process, but at large magnitudes. The authors should consider these two periods along with other fluctuations to generate a consistent interpretation. In any case, I don't think the proxies they use are adequate and robust enough to make convincing climate reconstructions.

Answer: Thank you very much for the comment. It could be seen from Fig.5 and Fig.6 (f)(g) in the manucript that the OM% and Rb/Sr ratio present opposite variations during the Holocene except the two deposition events. The Rb/Sr ratio increases with the decrease of OM% in peat sediment, but decreases synchronously with that of OM% in two deposition events. Therefore, it is reasonable to think that the mechanism of these two sedimentary events is different from that of the peat sediments. Moreover, some studies near our sampling site, such as Huang et al., 2015 and Zhao et al., 2016, have found tephra deposition in peat/lake sediment at the time intervals similar to these two sedimentary events. Therefore, we suspect that these two events may have been caused by volcanic events, too. However, we do not have enough evidence
to prove this conjecture, so we only focus on the results derived from the peat sediment in this paper and do not discuss these two events in detail. We'll conduct more precise sampling and experiments in our follow-up work to study these two events technically.

3.Despite that the authors identify existing problems and open questions about the Asian summer monsoon by citing many references, I don't think that the record presented here contribute much to the debate, due to the shortcoming of the proxy they use (see comments above). Also, they have to rely on previous work to distinguish temperature from precipitation changes, such as on page 5 lines 22-28, which also limit the new contribution from this study.

As such, I don't think that the large-scale climate discussions about insolation and solar forcing (Figs. 7 and 8) are supported by the evidence and arguments.

Answer: Thank you for the comment. Lake/peat records are usually multiinterpretative, and it is sometimes difficult to clarify a problem with a single proxy. Therefore, in lake/peat research, people usually use multiple proxies to investigate the climate changes. There have been many studies discussing the climate changes based on the proxy records derived from Hani peatland, but the results are quite different. Here we provide two new indicators, together with the proxy records from previous studies, in an attempt to reconcile these records and improve our knowledge of the climate change in this region. We think this work is meaningful and worth doing.

Furthermore, we discuss the connection between large-scale climate and the ice sheet/solar radiation not only based on our single record, but based on the synthesis between our own record and the records from northern China monsoon front area. The changes of these records are consistent and uniform since the last deglaciation, which probably represent the EASM variations, instead of regional changes. Therefore, these records could serve as an evidence of EASM variation around large area, rather than small region, which may not be robust, but should be reasonable.

4. The number of references is excessive. The authors cite >5 pages of references for

CPD
a 7-page manuscript! Most references are not needed.

In summary, based on the above considerations, I do not recommend the publication of the manuscript without additional analyses and improved proxy interpretations.

Answer: Thank you for the comment. In order to be precise, we have listed a lot of references to support our point of view, but it seems that they might be redundant. We'll delete a part as appropriate.

Specific comments:

1.Page 1 Line 26: insolation = incoming solar radiation, so "solar insolation" is redundant.

Answer: Thank you. We'll remove the "solar".

2.Line 26: "ka" often refers to 1000 year BP. Here it is better to say "kyr".

Answer: Thank you very much. We'll change "ka" to "kyr".

3.Page 2 Line 9: delete "." after "China"

Answer: Thank you. We'll delete it in the revision.

4.Line 17: I don't think Gorham (1991) is an appropriate reference for "studying past climate changes", as this is a seminal paper on peatland carbon stocks and their sensitivity to climate change, but not proxy/paleoclimate studies.

Answer: Thank you for the advice. We'll change it in the revised version.

5.Line 18-20: this excessive citation is not needed.

Answer: Thank you very much. The citations here were intended to illustrate the extensive use of peatlands in climate change. We'll remove some as appropriate.

6.Line 24-26: again too many references, which are unnecessary.

Answer: Thank you. We'll remove some as appropriate.

CPD
7.Line 29: I don't think "YD" has been defined (Younger Dryas). Define abbreviation when first used.

Answer: Thank you for the instruction. We'll define it firstly.

8.Page 3 Line 3: Hani peatland, not Hani peat.

Answer: Thank you. We'll revise it.

9.Line 4: probably use "peatland", rather than "swamp", as they are different. Swamp is too specific for woody peat-accumulating system with fluctuating water table.

Answer: Thank you for the advice. We'll change it to "peatland".

10.Line12: I'm not sure that these references are suitable for vegetation types, even though they may mention vegetation types, but likely were based on other sources. Also, move "." to the end of the sentence.

Answer: Thank you. We'll try to find the source of vegetation types or change to another references. And we'll move the "." to the end of the sentence.

11.Line 21-22: why bothering with dating pollen grains? The peat contains high OM, and you should either find identifiable macrofossil or bulk peat for dating. If worrying about potential organic material from aquatic plants (usually not abundant on that type of peatlands), then pollen grains could come from aquatic plants as well (unless you pick specific types of pollen grains, such as large pine or spruce pollen grains). I don't necessarily question the results, but it just appears to me the effort is unnecessary.

Answer: Thank you for the comment. There might be some micro organic carbon in the sediments, which could migrate up and down the water. We are afraid that if we use the bulk sediments for dating, these tiny content could probably effect the result. Therefore, we extracted the pollen grains for dating, hoping that the result would be better than using total organic matter.

12.Line 25-26: I think Bacon program would generate an age model, assigning age to
every depth, so I'm not sure you need a plotting program (OriginPro) for interpolation.

Answer: Thank you so much. The Bacon program could indeed assign age to every depth. However, we used the Bacon program to generate the age-depth plot after processing the AMS 14C data with Calib 7.0 and OriginPro 9.1. Therefore, even if it seems a bit redundant, we told about the process of data processing in the manuscript genuinely. In fact, there is no conflict between the two approaches.

13.Line 29: for LOI analysis, I'm not sure that weighted and grounded samples are necessary, unless grounded dry samples were also used for other analysis.

Answer: Thank you for the comment. The LOI550°C=(Ws-Wr)/Wsć100, in which "Ws" represents the dry weight of samples, "Wr" represents the weight of residue after burning. Therefore, in order to obtain the OM%, it is necessary to weight the samples. As for grinding, the samples are just ground to have no obvious lumps to ensure fully burned.

14.Page 4 Line 8. Delete "annual"

Answer: Thank you. We'll delete it.

15.Line 21-22: These two mineral layers should be discussed in order to understand other changes in OM% (that is, mineral content). See my general comment above.

Answer: Thank you. The reason why we didn't discuss the two mineral layers please see the interpretation above.

16.Line 12-15: this simplistic interpretation of LOI/OM is incorrect. No matter how productive is the peatland, if no external input of mineral materials, there will be no change in OM%. See my general comments above.

Answer: Thank you for the comment. We've found the pollen records of Hani, and the comparison between the pollen records and our OM% proves that the higher OM% possibly reflects a warm/humid climate. Please see our response above.
17.Line 18-21: This interpretation doesn't consider the transport process. Humid condition may cause great transport of fluvial-derived mineral materials, while dry condition may cause great source and transport of wind-blown mineral/dust. See my general comments above.

Answer: Thank you. Combining the OM and Rb/Sr ratio data with the pollen records, we considered the low Rb/Sr ratio in our peat sediment possibly corresponds to warm/humid climate condition. Please see the response above for details.

18.Page 7 Line 21: "major funding support"

Answer: Thank you for pointing out the mistake. We'll revise it.

19.Line 22: change "indoor" to "lab"?

Answer: Thank you for the advice. We'll change it.

20.Page 8 Too many references, and some of them are probably not necessary.

Answer: Thank you for the comment. We'll remove some as appropriate.

21.Page 14 Table 1. -indicate the unique dating lab ID number for each date, not just sample number. -indicate the depth range, such as 127-128 cm, or 126-127 cm?

Answer: Thank you for the advice. We'll modify them.

22. Figure 1 Detail map of study site is essential to understand the potential sources of mineral materials, through fluvial or eolian inputs.

Answer: Thank you. We'll add the satellite map of the study site in the revision (please see Fig.2).

Fig. 1.